# Quantifying Heterogeneity According to Deformation of the U937 Monocytes and U937-Differentiated Macrophages Using 3D Carbon Dielectrophoresis in Microfluidics

**DOI:** 10.3390/mi11060576

**Published:** 2020-06-08

**Authors:** Meltem Elitas, Esra Sengul

**Affiliations:** 1Faculty of Engineering and Natural Sciences, Sabanci University, Istanbul 34956, Turkey; sengulesra@sabanciuniv.edu; 2Sabanci University Nanotechnology Research and Application Center, Istanbul 34956, Turkey

**Keywords:** dielectrophoresis, deformation, mobility, heterogeneity, macrophage, monocyte

## Abstract

A variety of force fields have thus far been demonstrated to investigate electromechanical properties of cells in a microfluidic platform which, however, are mostly based on fluid shear stress and may potentially cause irreversible cell damage. This work presents dielectric movement and deformation measurements of U937 monocytes and U937-differentiated macrophages in a low conductive medium inside a 3D carbon electrode array. Here, monocytes exhibited a crossover frequency around 150 kHz and presented maximum deformation index at 400 kHz and minimum deformation index at 1 MHz frequencies at 20 V_peak-peak_. Although macrophages were differentiated from monocytes, their crossover frequency was lower than 50 kHz at 10 V_peak-peak_. The change of the deformation index for macrophages was more constant and lower than the monocyte cells. Both dielectric mobility and deformation spectra revealed significant differences between the dielectric responses of U937 monocytes and U937-differentiated macrophages, which share the same origin. This method can be used for label-free, specific, and sensitive single-cell characterization. Besides, damage of the cells by aggressive shear forces can, hence, be eliminated and cells can be used for downstream analysis. Our results showed that dielectric mobility and deformation have a great potential as an electromechanical biomarker to reliably characterize and distinguish differentiated cell populations from their progenitors.

## 1. Introduction

Dielectric parameters are among the essential biophysical properties of cells and can be associated with various immune and blood diseases [1,2,3,4,5]. Permeability and conductivity of the membrane and cytoplasm define dielectric properties of a cell in a specific microenvironment, which may change due to surface area of the cell as given by its size and shape; expression levels of surface proteins; form of cytoplasm; composition of cytos7ol; the surface charge density of the membrane; the morphologic complexity of membrane surfaces such as ruffles, microvilli, and blebs; as well as due to interfacial polarization of ions at the cell surfaces. Discovery of electrophysiological properties of cells, such as dielectrophoretic mobility, membrane relaxation period, crossover frequency difference, etc., relies on the phenomenon of dielectrophoresis (DEP), described by Herbert Pohl in 1951 [6]. Yet, intensive research has been conducted to utilize dielectrophoretic properties of cells to be label-free biomarkers for immune and blood diseases [7,8]. In this study, we interrogated whether dielectric movement and deformation measurements provide a specific, label-free, sensitive electromechanical biomarker for U937 monocytes and U937-differentiated macrophages.

Monocytes and macrophages can be considered as active machines that can immediately adapt to their microenvironment for pathogenesis and homeostasis through altering their electromechanical properties [9,10]. They are highly heterogenetic cells with their morphology, location, tissue-specific relations, and functional capabilities [11,12]. When examined by electron microscopy, the monocytes are spherical cells and they have microvilli and microcytotic vesicles, hence, their membrane surfaces have several ruffles and blebs, whereas macrophages have an irregular shape with electron-dense membrane-bound lysosomes. Besides, the microenvironment in which macrophages differentiate defines their shape, biochemistry and function [13]. Although we have been still investigating and discovering their new functions, such as the roles of macrophages in the electrical conduction of heart [14], in general, we know that monocytes enroll in tumor formation and invasion via metastasis and angiogenesis [15,16], macrophages are employed in pathogen recognition, phagocytosis [17], removal of dead cells and cellular debris [18] and tissue homeostasis [19,20]. Their diverse functions are continuously controlled by their dynamic microenvironment [21,22,23,24].

A pioneering work, sharing the purpose of determining electrical properties of mammalian cells according to their life cycle, was presented by Eisenberg and Doljanski in 1962. They measured the electrokinetic properties of liver cells in growth processes [25]. Next, Dr Petty’s research group reported heterogeneous distribution of electrophoretic mobilities of human monocyte subpopulations [26], while Dr Bauer and Dr Hannig determined the changes of the electrophoretic mobility (EM) of human monocytes during in vitro maturation into macrophages [27]. The current research direction, which investigates the change of cellular dielectrophoretic properties during the cell cycle, maturation or differentiation, mostly relies on determining the first crossover frequencies and measuring migration differences of cells [7,8,28,29]. Along the same lines, our previous investigations have presented the dielectrophoretic characterization and separation of U937 monocytes and U937-differentiated macrophages using their crossover frequencies and dielectrophoretic mobility differences according to their membrane permittivity and conductivity in a low conductive DEP buffer [30,31,32]. However, none of our previous studies have revealed dielectrophoresis-induced mechanical deformation of cells. Similarly, Tonin et al. interrogated electrophoretic mobility (EPM) during yeast growth and observed a nonmonotonic behavior during the cell cycle. They concluded that the maximal EPM occurred at the initial stage of the growth, and it strongly reduced at its final stage [33]. Song et al. employed DEP to sort human mesenchymal stem cells and their differentiation progeny, osteoblasts. Their results showed that osteoblasts experienced stronger DEP forces that laterally migrated them, whereas human mesenchymal stem cells remained on their original trajectories [34]. Dr. Salmanzadeh and his group used contactless DEP and observed that the trapping voltage of mouse ovarian surface epithelial cells increased as the cells progressed from a non-tumorigenic to a tumorigenic phenotype [35].

On the other hand, DEP has been utilized as a tool to stretch cells for characterization of their mechanical properties. It has provided great potential to implement single-cell biomechanical tests with high-throughput, automation, low complexity and cost, high scalability and portability in comparison to conventional biomechanical techniques, such as atomic force microscopy [36], optical tweezers [4,37], magnetic twisting cytometry [38], micropipette aspiration [39], diffraction phase microscopy [40] and microfluidic ektacytometry [41,42,43]. In this concept, Guido et al. demonstrated the capability of this new technique by characterizing deformability of cancerous MCF7 and noncancerous MCF10A cells [44]. Du and coworkers used this technique to reveal the biophysical properties of healthy, uninfected and infected red blood cells by Plasmodium falciparum malaria parasites [45].

In this study, we utilized dielectrophoresis to study the electromechanical properties of monocytes and macrophages that might quantify their population heterogeneity [11,45,46]. We measured the movement and calculated the deformation indexes of cells [47] under the influences of dielectrophoretic forces when 10–20 V_peak-to-peak_ (V_pp_) voltage with frequencies ranging from 50 kHz to 1 MHz have been applied.

## 2. Materials and Methods

### 2.1. DEP Buffer Preparation and Conductivity Measurement

DEP buffer with low electrical conductivity was prepared to keep cells viable during the processes of dielectrophoresis. As it has been previously reported [31], the low conductive DEP buffer [48] was composed of 8.6% sucrose (product no: LC-4469.1, NeoFroxx, Hesse, Germany), 0.3% glucose (CAS number 59-99-7, Sigma-Aldrich, Darmstadt, Germany) and 0.1% bovine serum albumin in distilled water (BSA, product code: P06-1391050, PAN-Biotech, Aidenbach, Germany).

The conductivity of the DEP buffer was 0.002 S/m, as measured by a Corning Model 311 Portable conductivity meter at room temperature (Cambridge Scientific Products, Watertown, MA, USA).

### 2.2. Cell Culture

In this study, U937 human monocyte cells (ATCC number: CRL1593.2) provided from ATCC (American Type Culture Collection, Manassas Virginia) and U937-differentiated macrophages were obtained by phorbol 12-myristate 13-acetate (PMA, Sigma Aldrich) treatment of U937 monocytes.

U937 cells were maintained in RPMI 1640 medium (Product Number: P04-18047, PAN-Biotech, Aidenbach, Germany) with 10% fetal bovine serum (FBS) (PAN Biotech, catalogue number: P40-37500, Aidenbach, Germany) using a T75 tissue culture flask (TPP^®^ Sigma, catalogue number: Z707554) at 37 °C with 5% CO_2_ in humidified air. U937 cells were grown until 80–90% confluency. Cells were centrifuged at 3000 rpm (Z601039, Hettich^®^ EBA 20 centrifuge, Merck, Darmstadt, Germany) for 5 min. The number of cells was determined using a hemocytometer (Marienfeld, Germany). The final cell concentration was adjusted to 3 × 10^6^ cells/mL.

The macrophage differentiation was performed using the 10 ng/mL concentration of the PMA treatment of 3 × 10^6^ U937 cells in 22.1 cm^2^ plates (TPP^®^ Product No:93060, Trasadingen, Switzerland) for 72 h. Next, the cells were maintained in medium without PMA for 48 h. Then, the cells were collected by treating with the 0.25% (v/v) Trypsin-EDTA (PAN Biotech, catalogue number: P10-019100, Aidenbach, Germany) solution. The cells were centrifuged at 1800 rpm (Z601039, Hettich^®^ EBA 20 centrifuge, Merck, Darmstadt, Germany) for 10 minutes to remove the remaining culture medium and washed twice using the DEP buffer.

### 2.3. 3D Carbon DEP Device

The fabrication process and features of the 3D carbon DEP devices were previously reported [48,49]. The carbon electrode array, a 1.8 mm wide, 3.2 cm long channel, was featured 218 intercalated rows with 14 or 15 electrodes each [50,51]. Individual electrodes had a height of 100 μm and a diameter of 50 μm (Figure 1). The numerical analysis to estimate the induced fluidic, electromagnetic and dielectrophoretic forces in the 3D carbon electrode array was earlier studied using both finite element analysis and numerical models [48,49,50,51].

### 2.4. Experimental Setup

The experimental setup consisted of a signal generator (Model: GFG-8216A, GW Instek, New Taipei City, Taiwan) with an oscilloscope (Part Number: 54622D, Agilent Technologies, Santa Clara, CA, USA) to create and observe the electric field, a desktop-acquired upright microscope (Model: Nikon ME600 Eclipse, Nikon Instruments Inc., Melville, NY, USA) to monitor cells and acquire images and a programmable syringe pump (Model: NE-1000, New Era Pump Systems Inc., Farmingdale, NY, USA) to flow the cells into the 3D carbon DEP device. We used 20–200 μL pipette tips (Manufacturer ID: 3120000917, Eppendorf, Hamburg, Germany) to connect microperforated Tygon tubing (Manufacturer ID: AAQ02103-CP S-54-HL, Cole-Parmer, Vernon Hills, IL, USA) into the inlet and outlet ports of the 3D carbon-DEP chip (Figure 1).

The experiment started with the sterilization of the electrode array using 70% ethanol and rinsing with deionized (DI) water using a syringe pump with a 20 µL/min flow rate. Next, the microfluidic chip was filled with the DEP buffer and the bubbles were removed. Then, 40 µL of the cell suspension was injected into the chip using a syringe pump with 10 µL/min flow rate. When the cells reached the electrode area, the flow was stopped, and the cells were released for 30 s. The experiments were started when the electric field was applied using the signal with 10–20 V_pp_ frequencies ranging from 50 kHz–1 MHz [30,31].

### 2.5. Image Acquisition and Data Analysis

The image sequences of cells were recorded using the Nikon ME600 Eclipse upright microscope (Nikon Instruments Inc., Melville, NY, USA) with 10× magnification in tiff sequence format. The VideoLAN Client (VLC, VideoLAN version 1.8, Paris, France) program was used to convert image sequences into the movies.

The acquired images were manually analyzed using open-access ImageJ software (Version 2.0 National Institutes of Health, Rockville, MD, USA). The crossover frequencies of single cells were determined by computing the movement of the cells according to their initial positions, as described in references [30,31]. In total, 50 monocyte cells and 30 macrophage cells were followed, and their positions were recorded. Using GraphPad Prism (Version 5.0) software, Student’s t-test was performed to compare dielectric mobilities of monocyte and macrophage populations. * implies that data are significantly different with *p* < 0.5.

The deformation index was calculated by manually measuring the height and width of 45 single monocyte and macrophage cells, and these single cells were continuously monitored, in each frequency. One-way analysis of variance and Tukey’s multiple comparison test were carried out using GraphPad Prism (Version 5.0) software to determine the significance. * and ** indicate that data are significantly different with *p* < 0.5 and *p* < 0.05, respectively. All measurements were provided in detail in the figure legends.

## 3. Results

### 3.1. Dielectrophoretic Movement

DEP offers the possibility to affect the movement of polarized particles in the non-uniform electric field. We can define the DEP force according to the difference between the dielectric properties of the particle and its suspension medium [52,53].
(1)$$F_{DEP}=2\pi r^{3}\varepsilon_{m}Re\left(K\left(\omega \right)\right)\nabla E^{2}$$

The DEP force ($F_{DEP}$) is related to the radius of the particle, the permittivity of the surrounding medium ($\varepsilon_{m}$), the real part of the Clausius–Mossotti factor (Re(K(ω)) and the applied electric field (E). The Clausius–Mossotti factor is defined as given by
(2)$$K\left(\omega \right)=\frac{\left(\varepsilon_{c}^{*}-\varepsilon_{m}^{*}\;\right)\;}{\left(\varepsilon_{c}^{*}+2\varepsilon_{m}^{*}\;\right)}$$

Here, $\varepsilon_{c}^{*}$ is known as the complex permittivity of a cell and $\varepsilon_{m}^{*}$ is the complex permittivity of the surrounding medium. The subscripts “*m*” and “*c*” mean suspending medium and cells, respectively. The complex permittivity can be expressed as
(3)$$\varepsilon^{*}=\varepsilon +\frac{j\sigma }{\omega }$$
where ε is the permittivity, σ is the conductivity and ω (ω=2πf) includes the electric field frequency. When the value of the Re(K(ω) is positive, the particle is attracted by the strong electric field region referred to as positive DEP (pDEP). When the value of the Re(K(ω) is negative, the particle is repelled by the high electric field region referred to as negative DEP (nDEP). The crossover frequency can be defined as the cessation of the particle motion, which is specific for the particles.

To quantify heterogeneity of monocytes and macrophages according to their dielectrophoretic behaviors, we applied the non-uniform AC electric field and determined the location of the cells in each frequency ranging from 50 kHz to 1 MHz (Figure 1). Our previous work presents the determination of the crossover frequencies in detail for the immune cells [30].

The translational movement of the cells was generated by dielectrophoretic forces and no fluid flow can introduce any drag force on the cells. Figure 2 demonstrates the number of cells that experienced strong pDEP (3), pDEP (2), weak pDEP (1), CF (0), weak nDEP (−1), nDEP (−2), strong nDEP (−3) at 50, 100, 200, 300, 400 and 1000 kHz frequencies when 20 and 10 V_pp_ voltages were applied for monocytes and macrophages, respectively.

Figure 2 demonstrates the dielectrophoretic behavior of the U937 monocytes and U937-differentiated macrophages under the influence of nonuniform electric field within the 3D carbon electrode array. Figure 2b shows that monocyte cells experienced nDEP to pDEP forces with increasing frequencies (n = 80 monocyte cells). The crossover frequencies of monocytes were between 100 to 200 kHz. The uniformity of pDEP responses of the monocytes was improved with increasing frequencies ranging from 200 kHz to 1 MHz, the strongest nDEP (−3, dark blue), the strongest pDEP (3, red) see Appendix A.

On the other hand, when the same experiment was performed using the U937-differentiated macrophage cells, they mostly exhibited pDEP behavior (warm colors yellow-red colors) and their weak crossover frequency was around 50 kHz (green), as shown in Figure 2b. The fraction of macrophage cells which immediately presented pDEP response was greater than the nDEP subpopulation. The number of nDEP experienced cells were not broadly changed in comparison to monocyte cells. Since most of the macrophage cells immediately experienced pDEP behavior and were attracted by the strong dielectrophoretic forces generated by 3D carbon electrodes, the number of analyzed cells in Figure 2b is limited to 30 cells; however, the initial number of cells was always 3 x× 10^6^ cells/ml for the experiments (see Materials and Methods Section 2.2. Cell culture, Appendix A).

The monocyte population showed smooth nDEP (blue) to crossover (green) and crossover to pDEP (red) transition as a whole monocyte population, as shown in Figure 2a. On the other hand, the macrophage population exhibited more likely a bimodal distribution that is either the macrophage cells in nDEP (blue) or pDEP (red) in comparison to the monocyte population (Figure 2b). Therefore, the dielectric movement of the U937-differentiated macrophages showed more heterogeneous population responses than the U937 monocyte population which is the originals of U937-differentiated macrophages.

Figure 3 compares the dielectrophoretic movement of the U937 monocytes and U937-differentiated macrophages. The macrophages moved from the nDEP region to pDEP region when 50 kHz at 10 V_pp_ was applied. The monocytes experienced nDEP to pDEP transition when 100–150 kHz at 20 V_pp_ was provided. When both the monocyte and macrophage populations exhibited strong pDEP forces at 1 MHz, there was not any significant difference between the trapping regions of the cells according to Student’s *t*-test (*p* value was 0.892, where * *p* < 0.5 was significant), as shown in Figure 3. This result may show that the interfacial polarization difference between the cytoplasm and plasma membrane can be stronger for macrophages than monocytes [54]. Therefore, the observed macrophage dielectric properties at 1 MHz can be related to both membrane and cytoplasm properties of macrophages, whereas the membrane features might dominate for the monocyte dielectric properties at 1 MHz. These varying biophysical properties between monocytes and macrophages might explain their distinct trapping regions inside the 3D carbon DEP device.

### 3.2. Dielectrophoretic Deformation Index

While dielectrophoretic forces distributed the cells in the electrode array according to their polarizability difference, DEP forces were also capable of creating deformation on the cells. As mentioned above, monocytes and macrophages are well-known cells for their plasticity properties [9,10]. When mammalian cells were exposed to large external flow forces in variable microenvironments using microfluidics, they became elongated, varied in size, and tended to return to their original shape once the external forces were removed [42,55].

We determined the dielectrophoretic deformation indexes (DDI) of the single U937 monocyte and the U937-differentiated macrophage cells using the non-uniform AC electric field varying from 50 kHz to 1 MHz frequency. The DDI values of each monocyte and macrophage cells were calculated for 47 cells as defined in Equation (4) [50], where *H* (µm) was the major and *W* (µm) was the minor axes of the cells, as shown in Figure 4a.
(4)$$DI=\frac{H}{W}$$

Figure 4 illustrated the DDI distribution for the monocytes (Figure 4b,d) and macrophages (Figure 4c,e), including the outliers. Monocyte population demonstrated significant DDI difference between 0–400 kHz, and 50–400 kHz at 20 V_pp_ (*p* < 0.5, Section 2 Materials and Methods, Section 2.5 Image acquisition and data analysis). The increased pDEP forces made the monocytes taller while attracting to the strong pDEP regions. When the pDEP forces reaches their maximum, the monocyte cells became wider and their deformation index significantly decreased at 300 kHz–1 MHz, and 400 kHz–1 MHz, 20 V_pp_ (*p* < 0.05), as shown in Figure 4b. Monocyte cells tended to generate pearl chain like organization under the influences of strong pDEP forces. Figure 4d displays the underlying dynamics of monocyte population when the change of deformation index was followed for each single cell. Single-cell analysis was performed when the DEP forces were applied for 50–500 kHz. The deformation index for the U937 monocytes were dynamically changed and created a zig-zag pattern within the 0.433–2.147 boundaries. Contrary to the deformation of monocytes, macrophages did not considerably alter their deformation (Figure 4c). Figure 4e demonstrates the deformation index of single macrophages that was exposed to DEP forces for the frequency range of 50–500 kHz. The change of deformation index for the U937-differentiated macrophage cells was more stable than U937 monocytes. The deformation indexes of macrophages exhibited smooth trajectories within the boundaries of 0.457–1.588.

Figure 5 demonstrates that there was a significant DDI difference between U937 monocytes and U937-differentiated macrophages at 300 kHz (*p* < 0.5) and 400 kHz (*p* < 0.05) according to Tukey’s multiple comparison test as explained in the Materials and Methods Section 2.5. (Image acquisition and data analysis). Monocyte population has higher DDI in comparison to macrophage population at 300 and 400 kHz, where both cell types were under the influences of pDEP forces. Next, increasing the frequencies decreased the DDI for monocyte cells, whereas it did not affect the DDI for macrophage cells. 

### 3.3. Dielectric Mobility and Membrane Relaxation Time

The principle of examining the polarized particles with DEP has been implemented to reveal the biophysical properties of cells since 1962 [25,33,56,57,58,59,60,61,62]. The strongest motivation beyond these studies has been the development of label-free dielectric biomarkers to distinguish healthy and pathological cells, since surface charge density of cells plays key roles in exocytosis, endocytosis, cell adhesion [63,64], binding of proteins [65,66,67] etc. The electrophoretic behavior of single cells has been predicted using the mathematical models that define the relationship between the mobility and the surface charges acting upon a cell suspending in a low conductive medium [67].

Here, we investigated whether dielectric mobility ($\mu_{DEP})\;$ [68] and membrane relaxation time (τ) [69] values are intrinsic, specific, dielectric markers that reliably distinguish U937 monocytes and U937-differentiated macrophages cell populations that have the same cell origin.

The dielectric mobility has been defined by Crowther and coworkers as in Equation (5), where η denotes the viscosity of the DEP buffer [68].
(5)$$\vec{v_{DEP}}=-\mu_{DEP}\nabla {\left|\vec{E}\right|}^{2}=-\left(\frac{\varepsilon_{m}r^{2}K\left(w\right)}{3\eta }\right)\nabla {\left|\vec{E}\right|}^{2}$$

The membrane relaxation time (τ) was expressed in Equation (6), where $C_{cell\;membrane}$ means the membrane capacitance of the cells [69].
(6)$$\tau =rC_{cell\;membrane}\left(\frac{1}{\sigma_{cell\;membrane}}+\frac{1}{2\sigma_{m}}\right)$$

Using the equations above, the dielectrophoretic mobility and membrane relaxation time values were calculated with the physical and electrical properties of the monocyte and macrophage cells, and the low conductive DEP buffer, as presented in Table 1.

The dielectric mobilities were calculated as 6.99 × 10^−18^ m^4^/V^2^s and 12.40 × 10^−18^ m^4^/V^2^s for monocytes ($\mu_{DEP_{Monocyte}}$) and macrophages ($\mu_{DEP_{Macrophage}})$, respectively. The membrane relaxation time values for the monocytes ($\tau_{Monocyte})\;$ were 2.63 × 10^5^ s, while $(\tau_{Macrophage})\;$ was 2.73 × 10^5^ s for the macrophages. Here, the membrane capacitance values used for the calculations were not belonged to specifically for the U937 monocytes and U937-differentiated macrophages, as noted in Table 1 [70,71,72]. To the best of our knowledge, the exact membrane capacitance value for the U937 macrophages has not been yet measured. Therefore, the values in Table 1 should be carefully interpreted.

## 4. Discussion

Monocytes and macrophage cells, sharing the same cell origin, have been compared according to their dielectrophoretic mobility and deformation. Both monocyte and macrophage populations exhibited inter-individual difference due to their intrinsic properties such as size, shape, and changes in membrane surface organization that may result in heterogeneity in their DEP responses.

Here, the crossover frequency of U937 monocytes was around 150 kHz. The U937-differentiated macrophage cells exhibited weak crossover frequency around 50 kHz (Figure 2). We used the computational tool for dielectric modeling published by Cottet, J. et al. and obtained the CM factor K(ω)  values for the monocytes ($K\left(\omega \right){\;}_{\mathrm{Monocyte}})$ and macrophages $\;\left(K\left(\omega \right){\;}_{\mathrm{Macrophage}}\right)$ as 0.976 and 0.979, respectively [67] (Table 1). Since the $K\left(\omega \right){\;}_{\mathrm{Macrophage}})$ was slightly higher than the ($K\left(\omega \right){\;}_{\mathrm{Monocyte}})$, macrophages were exhibited pDEP behavior earlier than monocytes (see Figure 2b,c and Figure 3). The uniformity of pDEP responses of the monocytes was improved with increasing frequencies (see Appendix A), the macrophages displayed both nDEP and pDEP fractions for the whole frequencies ranging from 50 kHz–1 MHz (see Appendix A). Although there was no significant difference between the trapping regions of the cells (Student’s t-test: *p* value was 0.892, where * *p* < 0.5 was significant), the DEP movement of macrophages were more heterogeneous than monocytes (Figure 3). We previously reported dielectrophoretic characterization and separation of U937 monocytes and U937-differentiated macrophages according to their crossover frequencies in [30,31,32].

This study, contrary to our previous work, reported that the translational DEP forces were not only moved cells according to their polarizability differences inside the electrode array, they also created irreversible deformation on the cells. Monocyte and macrophage cells display high plasticity among immune cells [9,10,42,55]. When DEP forces were introduced, the deformation index of monocytes first increased (0–400 kHz), then decreased with increasing pDEP forces (400 kHz–1 MHz), as shown in Figure 4. On the other hand, the deformation index of the macrophage cells did not exhibit significant difference for the frequencies ranging from 50 kHz to 1 MHz (Figure 4). When the dielectrophoretic deformation indexes of the monocyte and macrophage cell populations were compared, according to Tukey’s multiple comparison test, the increase in the deformation index of monocytes was significantly higher than the deformation index of macrophages at 300 kHz (*p* < 0.5) and 400 kHz (*p* < 0.05), as shown in Figure 5. Here, we calculated the DEP deformation indexes of the cells (Figure 4a: location of the cells according to electrodes) as we measured their translational mobility due to applied F_DEP_ (Figure 2a: position of the cells according to electrodes). Therefore, it relied on the spatial distribution of the cells within the electrode array since the DEP forces depend on polarizability of the cells according to their intrinsic properties. The applied DEP forces synchronized the cells spatiotemporally within the electrode array and we measured the deformation of single cells at their specific locations when the specific frequencies and voltages were applied, therefore, we achieved to obtain consistent results for the dielectric deformation indexes of the cells (Figure 4d,e).

In addition to experimental results, the dielectrophoretic mobility and membrane relaxation time values were predicted using the physical and electrical properties of the monocyte and macrophage cells, and the low conductive DEP buffer (Table 1) [68,69,70,71,72]. The calculated values were quite similar both for monocytes and macrophage cells. However, the values in Table 1 should be carefully interpreted since there are still unknown dielectric parameters for the U937 monocytes and U937-differentiated macrophages.

Our results marshalled considerable evidence for the feasibility of using dielectric mobility and dielectric deformation index as a dielectric biomarker that presents biophysical differences between the cell lines which shares the same origin. To the best of our knowledge, this is the first study that presents dielectric deformation indexes of cells and may become a practical method for achieving a specific, high-throughput, continuous, label-free, sensitive electromechanical characterization and classification technique for U937 monocytes and U937-differentiated macrophages.

Our further studies will focus on separation and recovery of cells with different deformation indexes from the 3D carbon DEP platform for downstream analysis using immunostaining and quantitative reverse transcription-polymerase chain reaction (RT-qPCR) techniques. Hence, we can promptly explain the dielectrophoretic mobility and deformation differences in terms of transcription and protein expression levels in the membrane surface and cytoskeletal components. Moreover, we can employ this method for further characterization of macrophage subpopulations, and it may provide value in increasing our understanding of the nature of tumor associated macrophages (TAMs).

## 5. Conclusions

This study presents heterogeneity of monocytes and macrophages according to their intrinsic dielectrophoretic properties in terms of dielectrophoretic deformation indexes. We performed dielectric deformation measurements of the U937 monocytes and U937-differentiated macrophages with similar radius and dielectric characteristics using 3D carbon electrode microfluidic platform both at population- and single-cell level. We calculated deformation indexes of the cells when 10–20 V_pp_ voltage with frequencies ranging from 50 kHz to 1 MHz have been applied.

Our results showed that the crossover frequency for the monocytes was around 150 kHz [30,31,32]. Monocytes presented maximum deformation at 400 kHz and minimum deformation around 1 MHz frequencies at 20 V_pp_. On the other hand, the crossover frequency for the macrophages, which differentiated from monocytes, was lower than 50 kHz, 10 V_pp_ [30,31,32]. Moreover, the dielectrophoretic deformation index for the macrophages was not significantly varied from 50 kHz to 1 MHz frequency range. We conclude that the change of the deformation index for macrophages was less in comparison to monocytes. Both dielectric mobility and deformation spectra revealed significant differences between the dielectric responses of U937 monocytes and U937-differentiated macrophages, which share the same origin.

Our method can be advanced for the development of label-free, specific, and sensitive single-cell characterization tools. This technique eliminates the possibility of damaging the cells by aggressive shear forces while allowing these cells to be used for further downstream analysis. To advance this work, we focus on development of automated image analysis tools to obtain directly deformation indexes and mobility data of cells from the acquired DEP videos.

Here, we particularly underlined F_DEP_-generated deformation index of monocytes and macrophages, since these cells are among the white blood cells which are capable of infiltrating different types of tissues. Further DEP studies might interrogate to quantify other immune cells or their subsets (TAMs), and whether their intrinsic cellular heterogeneity can be quantified according to their dielectrophoretic deformation indexes.

## Figures and Tables

**Figure 1 micromachines-11-00576-f001:**
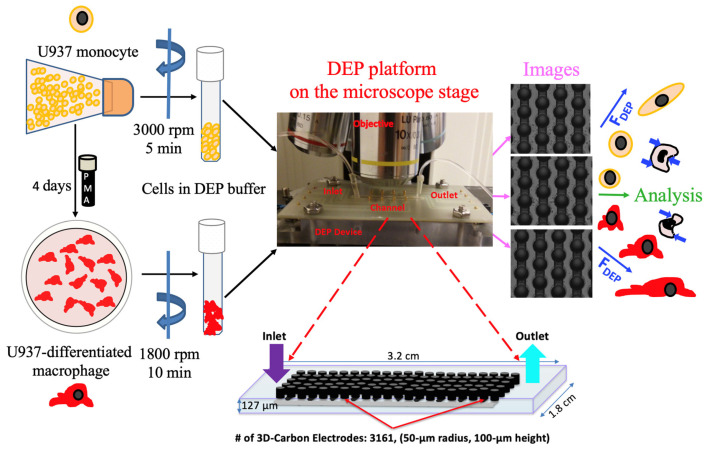
Schematic illustration of the experimental setup comprising the cell preparation step, 3D carbon electrode array, imaging and single-cell analysis.

**Figure 2 micromachines-11-00576-f002:**
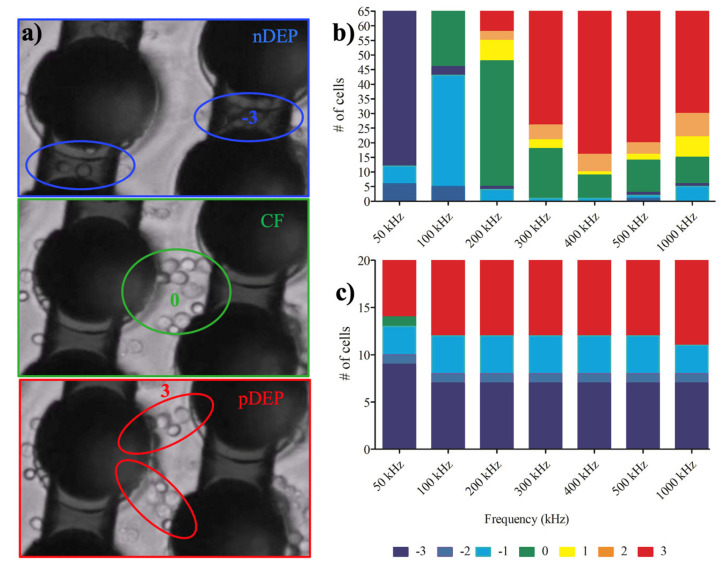
Dielectrophoretic responses of monocytes and macrophages: (**a**) Positions of the cells in the electrode array when they are influenced by nDEP, crossover frequency (CF) and pDEP, respectively; (**b**) Translational movement of U937 monocytes under 20 Vpp, 50 kHz–1 MHz nonuniform AC field; (**c**) Translational movement of U937-differentiated macrophages under 10 Vpp, 50 kHz–1 MHz nonuniform AC field. The cool colors show the number of nDEP- behaved cells due to repelling DEP forces while the warm colors demonstrate pDEP-responded cells owing to attractive DEP forces. Zero means the crossover frequency with zero movements, which is coded in green color. n = 80 for monocytes, n = 30 for macrophages.

**Figure 3 micromachines-11-00576-f003:**
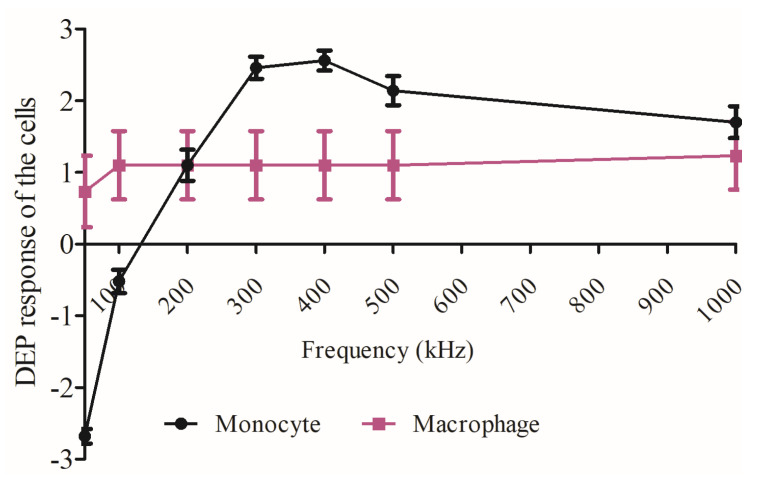
Comparison between the DEP movement of monocyte and macrophage cells. The magnitude of movement is categorized as very strong (3), strong (2), and weak forces (1). The “-” sign refers to nDEP. Measurements are the mean and error. n = 50 for monocytes, n = 30 for macrophages.

**Figure 4 micromachines-11-00576-f004:**
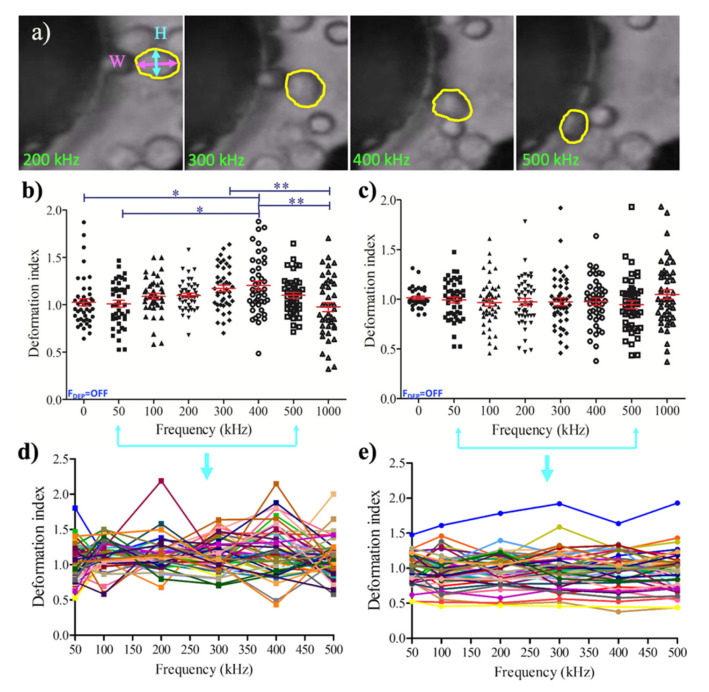
Dielectrophoretic deformation indexes (DDI) of U937 monocytes and U937-differentiated macrophages: (**a**) Representative image for the measurement of DDI. DDI values were presented with mean and standard error for population (n = 45). (**b**) single (n = 47) (**d**) monocyte cells; 45 population (**c**), single (n = 47) (**e**) macrophage cells. Tukey’s multiple comparison test is applied for (**b**). * and ** indicate that data are significantly different with *p* < 0.5 and *p* < 0.05, respectively. Each color displays the change of deformation indexes of single cells during the frequencies applied for the range of 50–500 kHz in (**d**) and (**e**).

**Figure 5 micromachines-11-00576-f005:**
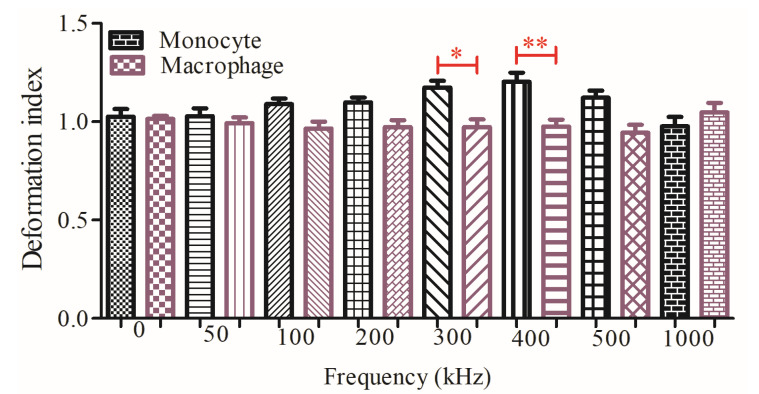
Comparison of the dielectrophoretic deformation indexes for the monocytes and macrophages without outliers. Measurements are the dielectrophoretic deformation index with mean and standard error for 45 monocyte and 45 macrophages cells. Tukey’s multiple comparison test is applied. * and ** indicate that data are significantly different with *p* < 0.5 and *p* < 0.05, respectively.

**Table 1 micromachines-11-00576-t001:** Dielectric markers specific to U937 monocytes and U937-differentiated macrophages.

Parameters (Units)	Values	Resources
$r_{Monocyte}$ (m)	1.15 × 10^−5^	Measured
$r_{Macrophage}$ (m)	1.5 × 10^−5^	Measured
$K\left(\omega \right){\;}_{Monocyte}$	0.976	Calculated [67]
$K\left(\omega \right){\;}_{Macrophage}$	0.979	Calculated [67]
$\varepsilon_{m}$ (C/V.m)	6.90 × 10^−10^	-
$\eta_{water}$ (kg/s.m)	8.90 × 10^−4^	-
$\sigma_{m}$ (S/m)	2 × 10^−3^	Measured
$\sigma_{Monocyte\;membrane}$ (S/m)	7 × 10^−13^	[70]
$\sigma_{Monocyte\;membrane}$ (S/m)	7 × 10^−13^	[70]
$\sigma_{Macrophage\;membrane}$ (S/m)	7 × 10^−13^	Assumed
$C_{Monocyte}$ (F/m^2^)	0.016 ± 0.002	[70,71]
$C_{Macrophage}$ (F/m^2^)	0.013 ± 0.001	[70,71]
$\tau_{Monocyte}$ (s)	2.63 × 10^5^	Calculated
$\tau_{Macrophage}$ (s)	2.73 × 10^5^	Calculated
$\mu_{DEP_{Monocyte}}$ (m^4^/V^2^s)	6.99 × 10^−18^	Calculated
$\mu_{DEP_{Macrophage}}$ (m^4^/V^2^s)	12.40 × 10^−18^	Calculated

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
