# Peer review of "Quantifying Heterogeneity According to Deformation of the U937 Monocytes and U937-Differentiated Macrophages Using 3D Carbon Dielectrophoresis in Microfluidics"

_micromachines, 2020, doi:10.3390/mi11060576_

Round 1

Reviewer 1 Report

This manuscript used the DEP to characterize the electromechanical properties of monocytes and macrophages. The cross-over frequency was obtained based on the switch from p-DEP to n-DEP. The deformation index was calculated based on the images of the cells subject to DEP force. This paper needs to address the following comments prior to be accepted:

  1. Since the DEP force depends on the location of the cell in the device, the degree of cell formation becomes spatially dependent, which might lead to different results on the deformation index. Authors need to discuss how to get consistent results based on cell deformation under DEP.
  2. It needs more data points to determine the cross over frequency. 
  3. It is better to use a simple mathematical model to estimate the cross-over frequency and then compare with the experimental results.    

Reviewer 2 Report

The manuscript "Single cell dielectrophoretic mobility and deformation; electromechanical biomarker to quantify population heterogeneity of monocytes and macrophages" reports a dielectrophoretic based microfluidic systems for measuring the deformation of U937 monocytes and U937-differentiated macrophages.   The work is very interesting, especially the use of 3D microelectrodes enabels a large number of cells to be investigated with the same device. I am happy to recommend the work to be accepted following a major revision, addressing my following comments:   1. Please revise the title, as your work does not report 'single cell' trapping or 'electromechanical biomarker' analysis, something like 'A 3D dielectrophoretic system for measuring the deformation of U937 monocytes and U937-differentiated macrophages in microfluidics' will be more suitable.   2. Please perform theoretical or numerical analysis to estimate the DEP force induced between the electrodes.   3. Please calculate the Clausius-Mossotti factor for U937 monocytes and U937-differentiated macrophages under various conditions reported in the paper, and relate it to Figures 2b-c and Figure 3.   4. The deformation index is a bit confusing, as it is defined as height to width ration but Figure 4a shows the ratio between major and minor axes of the cells. Please clarify this.   5. If dielectrophoresis changes the deformation index of the cells, you need to show cells before and after exposure to dielectrophoresis so that the reader can see the difference. Please also discuss the relationship between deformation index and the dielectrophoretic force.   6. Please discuss the risk of localised temperature peaks and electrothermal vortices due to application of electrical field, as presented in: Analytical Chemistry, 2011, 83 (6), 2133-2144   7. Please discuss and properly cite the previous work showing the interfacing of U937 monocytes with dielectrophoresis, as presented in Analytical Chemistry, 2011 83 (8), 3217-3221, and the benefits offered by your proposed system.   8. Please discuss opportunities for coupling your system to inertial microfluidic systems, as reported in Lab on a Chip, 2019, 19 (17), 2885-2896 to immobilize target cells following being released from the inertial system.

Round 2

Reviewer 2 Report

I am happy to recommend the manuscript to be accepted in Micromachines.